# Thermodynamic Analysis and Crystallographic Properties of MFe_2_O_4_, MCr_2_O_4_ and MAl_2_O_4_ (M = Fe, Ni, Zn) Formed on Structural Materials in Pressurized Water Reactor Primary Circuit under Zinc and Zinc-aluminum Water Chemistry

**DOI:** 10.3390/e24020245

**Published:** 2022-02-06

**Authors:** Yang Jiao, Shenghan Zhang, Yu Tan

**Affiliations:** 1Hebei Key Lab of Power Plant Flue Gas Multi-Pollutants Control, Department of Environmental Science and Engineering, North China Electric Power University, Baoding 071003, China; 1182102069@ncepu.edu.cn (Y.J.); lucifertan@163.com (Y.T.); 2Environmental System Optimization, College of Environmental Science and Engineering, North China Electric Power University, Beijing 102206, China

**Keywords:** thermodynamic property, solubility, zinc injection, zinc-aluminium simultaneous injection, oxide film, PWR primary circuit

## Abstract

Zinc injection technology (zinc water chemistry, ZWC) was widely applied in pressurized water reactor (PWR) primary circuits to reduce radiation buildup and improve corrosion resistance of structural materials. The simultaneous injection of zinc-aluminium (ZAWC) is a novel implement created to replace part of Zn^2+^ by Al^3+^. It was reported ZAWC can improve further corrosion resistance of carbon steels and stainless steels. However, ZAWC sometimes showed even negative effects on Nickel-alloys. In this study, mechanism of formation of oxide film on metals was investigated. The reactions of Fe^2+^ Ni^2+^ in oxide films replaced by Zn^2+^, or Fe^3+^ replaced by Al^3+^ in ZAWC were analysed. The thermodynamic data and solubility of mixed oxides (ZnFe_2_O_4_, ZnCr_2_O_4_, and ZnAl_2_O_4_), the products of replace reactions, were calculated. According to the Gibbs free energy difference between products and reactants, $\Delta G^{\theta }\left(T\right)$ values of the formation reaction of ZnFe_2_O_4_, ZnCr_2_O_4_, and ZnAl_2_O_4_ are extremely negative. Solubility of ZnAl_2_O_4_ is the lowest among mixed oxide products, which implies the oxide film composites of ZnAl_2_O_4_ may show a lower corrosion rate. In addition, the preferential formation of NiAl_2_O_4_ on Ni-based-alloy, under ZAWC, was discussed based on crystallographic properties of spinel, which was considered as the cause of negative effects of ZAWC on corrosion resistance of Nickel-alloys. This research provides an analytical basis for the study of thermodynamic stability of oxide films under different chemical chemistry and a theoretical basis for improving corrosion resistance of different metals and optimizing the chemical conditions of PWR primary circuit.

## 1. Introduction

Light water reactors (LWR) and pressurized heavy water reactors (PHWR) are the main reactor types of nuclear power plants. LWR are divided into pressurized water reactors and boiling water reactors (BWR). The main materials in PWR primary system (Figure 1) are composed of 25% zirconium alloys as fuel cladding, 65% Ni-based alloys as steam generator (SG) tubes, 5% stainless steels (SS) as SG, and some carbon steels (CS). At the same time, stainless steels are also the main materials of piping in PWR primary circuit [1,2].

The component materials in PWR primary circuit are listed in Table 1. Based on PWR primary coolant water chemistry conditions, it has been reported that iron oxides are the main component in oxide films formed on carbon steels. It was also well known that oxide films, formed on stainless and nickel-based alloys in high temperature water, had a double-layer structure: a (Ni, Fe)-rich outer layer and a Cr-rich inner layer [3,4,5,6].

The corrosion and radiation buildup reduction effects of zinc injection into PWR primary coolant has been experimentally and practically confirmed in past years [8,9,10,11]. As the improvement plan of zinc injection technology, the inhibitory effects of simultaneous injection of zinc and aluminium, on the general corrosion of different metal materials, has also been proved in the previous studies in recent years. However, the influence on different materials were different. Zhang et al. [12] claimed that the corrosion resistance of oxide films, formed on 316L SS in this new technology, was significantly improved compared than zinc injection. In the research of Sun [13], it has been pointed out that the effects of ZAWC on the corrosion behaviour of carbon steel, stainless steel, and Ni-based alloy were different. Compared with conventional water chemistry, the corrosion current density of oxide film formed on A508-3 CS reduced by 30% under ZWC, but almost 75% for 304L SS. After zinc-aluminium was simultaneously injected into boron-lithium solution, the corrosion resistance of CS and SS was significantly enhanced, and the corrosion current density could be reduced by an order of magnitude, but the improvement effect was weak for Ni-based alloys. The specific effects of these two implements on the corrosion current density were shown in Figure 2, in which “PWR” represents the common PWR coolant water chemistry, and, assuming that the corrosion current density in this environment was 100%, the percentage shown in the vertical axis was relative to the value of PWR.

The previous results were obtained, mainly, through experiments, especially for the condition of ZAWC. However, no detailed mechanism analysis was carried out for the explanation of differences between different types of metal materials. The consideration of oxide composition under high temperature aqueous conditions is fundamental to the corrosion behaviour and stability of alloys. In order to evaluate the effectiveness of zinc and zinc-aluminium injection on the corrosion behaviour of carbon steels, stainless steels, and Ni-based alloys, thermodynamic data and solubility calculation at high temperatures are carried out to compare the thermal stability of oxide films formed on metal surface. Based on the Laws of Thermodynamics, the lower value of Gibbs free energy at high temperatures means the high thermodynamic stability of substance. When the Gibbs free energy difference between products and reactants of reactions is less than zero, it indicates that the reaction between oxide film and solution can proceed spontaneously, and the more negative the value, the higher degree of spontaneity. Moreover, the parameters and crystallographic features of spinel are suggested to illustrate the preference of different cation ions, which can judge the structural stability of spinel oxides formed on the surface of metals.

## 2. Thermodynamic

The composition of oxide films formed on different materials in simulated PWR primary coolant have been confirmed in previous researches. The effect of the injection of different ions into the coolant of PWR primary circuit on the oxide film on the metal surface and the mechanism diagram are shown in Figure 3. In PWR primary circuit, passive films with large polyhedral oxides form on carbon steels. Generally, the oxides are mainly composites of iron oxides and minor other metal oxides. After Zn^2+^ added into primary circuit, Zn^2+^ was incorporated into the oxide films, and finally, small-size polyhedral oxides (mixed oxides of iron and zinc) formed on the surface. However, due to the low concentration of Zn^2+^, a number of large polyhedral also exist. On the condition of Zn^2+^ and Al^3+^ simultaneous injection, Al^3+^ may enter the oxide film through substitution reactions. Generally, oxide films, formed on stainless steels and Ni-based alloys in high temperature water, have a duplex structure with a Cr-rich inner layer and Cr-depleted outer layer. Under ZWC and ZAWC conditions, Zn^2+^ and Al^3+^ can also participate in the reactions to form smaller and denser protective oxides.

The specific analysis and the thermodynamic calculation of the formation reaction of various substances are studied in this chapter.

### 2.1. Zinc Injection

It has been generally acknowledged that iron oxides (Fe_3_O_4_, Fe_2_O_3_, FeO…) were the main oxides on carbon steels that protect the surface from corrosion. When dozens of ppb Zn^2+^ are added to primary coolant, ZnFe_2_O_4_ may be generated by Reaction (1):(1)$${\mathrm{Fe}}_{3}O_{4}{\;+\;\mathrm{Zn}}^{2+}\to {\mathrm{ZnFe}}_{2}O_{4}{\;+\;\mathrm{Fe}}^{2+}$$

The different oxide structures and composition were observed when water chemistry (whether hydrogen or oxygen in solution) and materials were in difference. Fe_3_O_4_-FeCr_2_O_4_ double-layer oxide films were formed on the surface of 304 stainless steel when exposed to simulated primary water with hydrogen [11,14]. Because Ni percentage of 316L SS is a little higher than that in 304L SS (Table 1), the generation of Ni_x_Fe_1−x_O_4_ in outer layer and Ni_1−x_Fe_x_Cr_2_O_4_ in inner oxide were also taken into consideration on the surface of stainless steel [15]. All in all, the particles of oxide film in outer layer were large, loose and uneven in size, while the inner oxides were small size and uniform [16,17]. Under Zn injection condition, ZnFe_2_O_4_ and ZnCr_2_O_4_ were generated as the main composition of oxide film on stainless steels. However, it has been reported by Liu et al. that, after a long-term oxidation, ZnCr_2_O_4_ became the dominant corrosion oxide. The possible main reaction equations were (1) as above and (2) (3) (4) as follows. It was generally acknowledged that the formation of the dense Zn_x_Fe_1−x_Cr_2_O_4_ inner layer was the main reason for the improvement of corrosion resistance under a high temperature and high pressure water environment. Moreover, the replacement of Fe^2+^ by Zn^2+^ should be a no mixing limitation, but the ration of substitution depends on the concentration of Zn^2+^ and immersion time in the Zn-containing solution [11].
(2)$${\mathrm{NiFe}}_{2}O_{4}{\;+\;\mathrm{Zn}}^{2+}\to {\mathrm{ZnFe}}_{2}O_{4}{\;+\;\mathrm{Ni}}^{2+}$$
(3)$${\mathrm{FeCr}}_{2}O_{4}{\;+\;\mathrm{Zn}}^{2+}\to {\mathrm{ZnCr}}_{2}O_{4}{\;+\;\mathrm{Fe}}^{2+}$$
(4)$${\mathrm{NiCr}}_{2}O_{4}{\;+\;\mathrm{Zn}}^{2+}\to {\mathrm{ZnCr}}_{2}O_{4}{\;+\;\mathrm{Ni}}^{2+}$$

After long-term exposure to PWR high temperature coolants, oxide films with double-layer stable structure has also been found to be formed on the surface of nickel-based alloys (such as Alloy 600, Alloy 690, and Incoloy 800). It was believed that the selective release process of nickel resulted in the formation of the chromium-rich inner oxide layers. Some of the released nickel cations will be precipitated to form (Fe, Ni)-rich outer oxide particles, while others will be transported to, and activated in, the reactor core to increase the radiation fields [18]. Nickel-based alloy has high percentage of Ni in the matrix, and as a result, both outer and inner oxide are composed of Ni-rich oxide. It has been verified that Fe_x_Ni_1−x_Fe_2_O_4_ inverse spinel was the main composition of outer layer and Fe_x_Ni_1−x_Cr_2_O_4_ for inner layer. Similarly, ZnFe_2_O_4_ and ZnCr_2_O_4_ were expected to generate on Ni-based alloys with zinc injection by reaction (2) and (4). As shown in previous research, outer oxide particles were not densely packed but rather were unevenly dispersed.

In general, after zinc injection into PWR primary coolant, Zn^2+^ first enters the outer layer oxide and finally, to the inner layer oxide film. After the inner layer is saturated, a dynamic equilibrium will be established between the outer layer oxide and zinc in the solution. Therefore, it is necessary to calculate the thermodynamic data of Fe_3_O_4_, NiFe_2_O_4_, FeCr_2_O_4_, NiCr_2_O_4_, ZnFe_2_O_4_, and ZnCr_2_O_4_ and investigate the generation of each oxidation product under PWR primary circuit conditions.

The Gibbs free energy ${\Delta G}_{f}^{\circ }\left(T\right)$ of different substances and ions, involved in the calculation process at different temperature T, can be calculated by Equation (5):(5)$${\Delta G}_{f}^{\circ }\left(T\right){=\Delta G}_{f}^{\circ }\left(298K\right)-\left(T-298.15\right){{}^{\circ}\times {}^{\circ}S}_{298}^{{}^{\circ}}+\int_{298}^{T}C_{P}\mathrm{dT}-T\int_{298}^{T}\frac{C_{P}}{T}\mathrm{dT}$$where ${\Delta G}_{f}^{\circ }\left(298K\right)$ represents the Gibbs free energy at 298K, $S_{298}^{\;}$ is the entropy value at 298K, and $C_{P}$ is the heat capacity value, which can be calculated by Equation (6):(6)$$C_{P}{=A+\mathrm{BT}+\mathrm{CT}}^{-2}$$where A, B, C are the heat capacity constants of different substances. All the thermodynamic data of species used in formation equations, obtained with the aforementioned method, are listed in Table 2.

The Gibbs free energy ${\Delta G}_{f}^{\circ }\left(T\right)$ of each species at high temperatures from 473K (200 °C) to 623K (350 °C), as simulated with PWR primary conditions, are calculated by Equation (5) and listed in Table 3.

As the temperature rises from 473K to 623K, ${\Delta G}_{f}^{\circ }\left(T\right)$ of spinel oxides gradually decreases. Moreover, it is worth mentioning that ${\Delta G}_{f}^{\circ }\left(T\right)$ of ZnCr_2_O_4_ is lower than other substances formed on the surface of alloys, which directly indicates the high stability of ZnCr_2_O_4_ in aqueous systems at high temperatures.

According to the basic thermodynamic theory, when the Gibbs free energy change ${\Delta G}^{\theta }\;(T)$ is less than 0 under certain conditions, the reaction can proceed spontaneously. With zinc injection technology applied to PWR primary circuit, the double-layer oxide film composed of ZnFe_2_O_4_ and ZnCr_2_O_4_ are expected to be generated on the surface of alloys. The ${\Delta G}^{\theta }\;(T)$ of possible formation reactions of oxidation products are calculated by Equation (7) and shown in Figure 4 as a function of temperature.
(7)$${\Delta G}^{\theta }{(T)=\Sigma v}_{i}{\Delta G}_{f}^{\circ }\left(\mathrm{Product}\right){-\Sigma v}_{i}{\Delta G}_{f}^{\circ }\left(\mathrm{Reactant}\right)$$

It can be seen in Figure 4 that, when temperature is higher than 500K, ${\Delta G}^{\theta \;}\left(T\right)$ of Reaction (2) (3) and (3) are negative, indicating the generation reaction can proceed spontaneously. As the temperature increases, the value of ${\Delta G}^{\theta \;}\left(T\right)$ decreases, which means the reactions are easier to occur in high temperature environments. Moreover, ${\Delta G}^{\theta \;}\left(T\right)$ of Reaction (4) behaves with the lowest value among these changes. From the thermodynamic point of view, the substitution reaction of NiCr_2_O_4_ (the replacement of Ni^2+^ by Zn^2+^) is more likely to proceed under PWR primary circuit condition. The Gibbs free energy changes of Reaction (1) are positive, showing that this reaction cannot react spontaneously under the calculated condition. In consequence, the formation of ZnFe_2_O_4_ from Fe_3_O_4_ is analysed through the crystallographic properties of spinel.

### 2.2. Zinc-Aluminum Simultaneous Injection

Compared with zinc injection technology, Al^3+^ injection, alone, had a weaker corrosion inhibition effect. However, previous studies have shown that aluminium ion, injected into simulated PWR primary coolant, can effectively prevent the diffusion and deposition of Co^2+^, enhance the corrosion activation energy of stainless steel, and solve the problem of cumulative radiation [19]. Therefore, the experiments on the replacement of part of zinc by aluminium into PWR primary circuit water (that was called zinc-aluminium simultaneous injection (ZAWC)) were carried out. The results have proven that this new implement can further improve the corrosion resistance of metal materials on the basis of zinc injection in PWR primary circuit [20,21].

The formation of new phases, ZnAl_2_O_4_, FeAl_2_O_4_, and NiAl_2_O_4_, with high protectiveness in the passive film on the surface of carbon steel, stainless, and nickel-based alloys are the main reason for the enhancement of corrosion resistance. When zinc and aluminium are simultaneously injected, the following reactions may occur between the passive films and solution, in which (8)–(13) may be the reactions in outer layer and (14)–(16) for inner layer.
(8)$${\mathrm{Fe}}_{3}O_{4}{\;+\;\mathrm{Zn}}^{2\;+\;}{\;+\;2\mathrm{Al}}^{3\;+\;}\to {\mathrm{ZnAl}}_{2}O_{4}{\;+\;2\mathrm{Fe}}^{3\;+\;}{\;+\;\mathrm{Fe}}^{2\;+\;}\;$$
(9)$${\mathrm{NiFe}}_{2}O_{4}{\;+\;\mathrm{Zn}}^{2\;+\;}{\;+\;2\mathrm{Al}}^{3\;+\;}\to {\mathrm{ZnAl}}_{2}O_{4}{\;+\;2\mathrm{Fe}}^{3\;+\;}{\;+\;\mathrm{Ni}}^{2\;+\;}$$
(10)$${\mathrm{Fe}}_{2}O_{3}{\;+\;\mathrm{ZnO}\;+\;2\mathrm{Al}}^{3\;+\;}\to {\mathrm{ZnAl}}_{2}O_{4}{\;+\;2\mathrm{Fe}}^{3\;+\;}$$
(11)$${\mathrm{ZnFe}}_{2}O_{4}{\;+\;2\mathrm{Al}}^{3\;+\;}\to {\mathrm{ZnAl}}_{2}O_{4}{\;+\;2\mathrm{Fe}}^{3\;+\;}$$
(12)$${\mathrm{Fe}}_{3}O_{4}{\;+\;\mathrm{Al}}^{3\;+\;}\to {\mathrm{FeAl}}_{2}O_{4}{\;+\;\mathrm{Fe}}^{3\;+\;}$$
(13)$${\mathrm{NiFe}}_{2}O_{4}{\;+\;\mathrm{Al}}^{3\;+\;}\to {\mathrm{NiAl}}_{2}O_{4}{\;+\;\mathrm{Fe}}^{3\;+\;}$$
(14)$${\mathrm{FeCr}}_{2}O_{4}{\;+\;\mathrm{Al}}^{3\;+\;}\to {\mathrm{FeAl}}_{2}O_{4}{\;+\;\mathrm{Cr}}^{3\;+\;}$$
(15)$${\mathrm{NiCr}}_{2}O_{4}{\;+\;\mathrm{Al}}^{3\;+\;}\to {\mathrm{NiAl}}_{2}O_{4}{\;+\;\mathrm{Cr}}^{3\;+\;}$$
(16)$${\mathrm{ZnCr}}_{2}O_{4}{\;+\;\mathrm{Al}}^{3\;+\;}\to {\mathrm{ZnAl}}_{2}O_{4}{\;+\;\mathrm{Cr}}^{3\;+\;}$$

The calculation process is the same as Section 2.1. The thermodynamic data and ${\Delta G}_{f}^{\circ }\left(T\right)$ at different temperatures are listed in Table 4 and Table 5.

The Gibbs free energy changes of Reaction (8)–(16) are calculated by Equation (7) at different temperatures, from 473K to 623K, and the data is shown in Figure 5.

Firstly, the Gibbs free energy ${\Delta G}_{f}^{\circ }\left(T\right)$ of ZnAl_2_O_4_ is the lowest among oxides formed on alloys (Table 5), indicating the highest stability of this spinel oxide. Secondly, it can be seen that the Gibbs free energy changes in Figure 5a are significantly lower than (b), which means the high spontaneity of reaction of the formation of ZnAl_2_O_4_.

## 3. Solubility

The formation reactions of oxide films has been discussed above from the perspective of thermodynamics. In this chapter, the solubility of each substance in double-layer oxides are calculated to further investigate the stability of passive films formed on alloys, under ZWC and ZAWC conditions, at high temperatures. Moreover, the influences of zinc and zinc-aluminium injection on the corrosion resistance of different metals are also analysed.

### 3.1. Cr-Rich Inner Laryer Oxide

The solubility of FeCr_2_O_4_, NiCr_2_O_4_, and ZnCr_2_O_4_, formed in the inner layer oxide, are calculated using the method from K. Miyajima et al. [2]. Electrochemical process of spinel oxide M in aqueous systems can be expressed by Reaction (17), and the dissolution of solid M to produce ions in solution N can be written by general Reaction (18):(17)$${M+\alpha H}^{+}{+\beta e}^{-}=\gamma N+\frac{\alpha }{2}H_{2}O$$
(18)$$M+\left(\alpha -\beta \right)H^{+}+\frac{\beta }{2}H_{2}=\gamma N+\frac{\alpha }{2}H_{2}O$$

The possible dissolution equilibrium reactions of FeCr_2_O_4_, NiCr_2_O_4_, and ZnCr_2_O_4_ are shown in Table 6 in the form of Reaction (18).

The thermodynamic data of each substance and ion are listed in Table 7, and the value of formation free energy changes ${\Delta G}^{\theta \;}\left(T\right)$ are calculated using Equation (7). The equilibrium constant can be used to calculate the activity of N ($\alpha_{N}$). For low ionic strength solution, the activity coefficient is equal to 1, so $\alpha_{N}$ is equal to the concentration of ion N ([N]) and can be calculated by Equation (19):(19)$$\log_{10}\alpha_{N}=-\frac{{\Delta G}^{\theta }\left(T\right)}{2.303\mathrm{RT}\gamma }-\frac{\alpha -\beta }{\gamma }{\mathrm{pH}}_{T}+\frac{\beta }{2\gamma }\log_{10}P_{H_{2}}$$
where the value of hydrogen partial pressure $P_{H_{2}}=0.1\;\mathrm{atm}$ [19]. The total solubility of spinel oxide M is equal to the sum of ionic concentrations from the reactions, which contribute to the dissolution of M. Figure 6 shows the solubility of chromium-rich oxide formed in the inner layer at different temperature as a function of pH.

It can be generally seen, in Figure 6, that with the increase in pH, the solubility of Cr-rich oxides show a trend of first decreasing and then increasing. For FeCr_2_O_4_ and ZnCr_2_O_4_, with the increase in temperature from 473K to 623K, their solubility increases, indicating that the dissolution of these oxides are endothermic reaction. The lowest value displays at about pH = 6 at 473 K. As the temperature increases, the turning point shifts to the right side. However, the turning point of NiCr_2_O_4_ solubility occurs in the alkaline environment. When the pH is in the range of 7–8, the value of solubility behaves at a relatively low level.

The hydrogen partial pressure ($P_{H_{2}}$) in Equation (19) is necessary to calculate the solubility of oxides, so the difference between various studies were compared in Figure 6c. The hydrogen partial pressure of 0.252 atm was used in the calculation of K. Miyajima’s research [2]. Due to the higher $P_{H_{2}}$ value than this article, the solubility of ZnCr_2_O_4_ was relatively high, shown with short dots. However, the assumed $P_{H_{2}}$ was in the range of 0.0019–0.1134 atm at 573.15K in Ref. [11], which was similar to the calculation above. Although the inflection point varies between references and this study, which may be due to the different ions and dissolution equilibrium reactions considered in the calculation, the overall trend of solubility was similar. When pH is higher than 6, especially in the pH range of PWR primary circuit coolant at 6.9–7.3, the value of solubility, it is very close to the calculated value (shown with dash line). It can be justified that the calculation results shown above were relatively credible.

### 3.2. Fe-Rich Outer Layer Oxide

The calculation method used for the solubility outer layer oxide Fe_3_O_4_, NiFe_2_O_4_, and ZnFe_2_O_4_ is from Shi et al. [20,26]. Firstly, it is assumed that Fe_3_O_4_/NiFe_2_O_4_ and Fe_2_O_3_ coexist in aqueous solution. After zinc injection is implanted, the dissolution reactions are based on the assumption of coexistence of ZnFe_2_O_4_ and Fe(OH)_3_. The possible dissolution reactions are listed in Table 8, and the reaction equilibrium constant K can be calculated by Equation (20) according to the thermodynamic data shown in Table 9.
(20)$${K=e}^{-\frac{\Delta G^{{}^{\circ}}\left(T\right)}{\mathrm{RT}}}$$

The solubility of Fe-rich outer layer oxide is the sum of concentration of every ion and substance. Thus, the calculation equation can be expressed by Equations (21)–(28). In Equations (14)–(28), M represents the element of Fe, Ni, and Zn. Figure 7 demonstrates the solubility of Fe_3_O_4_, NiFe_2_O_4_, and ZnFe_2_O_4_, formed in the outer layer at different temperature, as a function of pH.
(21)$$S_{{\mathrm{Fe}}_{3}O_{4}}=\left[{\mathrm{Fe}}^{2+}\right]+\left[\mathrm{Fe}{\left(\mathrm{OH}\right)}^{+}\right]+\left[\mathrm{Fe}{\left(\mathrm{OH}\right)}_{2}^{0}\right]+\left[{\mathrm{Fe}(\mathrm{OH})}_{3}^{-}\right]$$
(22)$$S_{{\mathrm{NiFe}}_{2}O_{4}}=\left[{\mathrm{Ni}}^{2+}\right]+\left[\mathrm{Ni}{\left(\mathrm{OH}\right)}^{+}\right]+\left[\mathrm{Ni}{\left(\mathrm{OH}\right)}_{2}^{0}\right]+\left[{\mathrm{Ni}(\mathrm{OH})}_{3}^{-}\right]+\left[{\mathrm{Ni}(\mathrm{OH})}_{4}^{2-}\right]$$
(23)$$S_{{\mathrm{ZnFe}}_{2}O_{4}}=\left[{\mathrm{Zn}}^{2+}\right]+\left[\mathrm{Zn}{\left(\mathrm{OH}\right)}^{+}\right]+\left[\mathrm{Zn}{\left(\mathrm{OH}\right)}_{2}^{0}\right]+\left[{\mathrm{Zn}(\mathrm{OH})}_{3}^{-}\right]+\left[{\mathrm{Zn}(\mathrm{OH})}_{4}^{2-}\right]\;$$
(24)$$\left[M^{2+}\right]{=e}^{-\frac{\Delta G_{1}^{{}^{\circ}}}{\mathrm{RT}}-\left(2\ln10\right)\mathrm{pH}}$$
(25)$$\left[M{\left(\mathrm{OH}\right)}^{+}\right]{=e}^{-\frac{\Delta G_{2}^{{}^{\circ}}}{\mathrm{RT}}-\left(\ln10\right)\mathrm{pH}}$$
(26)$$\left[M{\left(\mathrm{OH}\right)}_{2}^{0}\right]{=e}^{-\frac{\Delta G_{3}^{{}^{\circ}}}{\mathrm{RT}}}$$
(27)$$\left[{M(\mathrm{OH})}_{3}^{-}\right]{=e}^{-\frac{\Delta G_{4}^{{}^{\circ}}}{\mathrm{RT}}+\left(\ln10\right)\mathrm{pH}}$$
(28)$$\left[{M(\mathrm{OH})}_{4}^{2-}\right]{=e}^{-\frac{\Delta G_{5}^{{}^{\circ}}}{\mathrm{RT}}+\left(2\ln10\right)\mathrm{pH}}$$

The solubility of Fe-rich oxides shows the same trend as Cr-rich oxides in Section 3.1 in that it decreases first and increases after pH is higher than 9. S_Fe3O4_ increases greatly under the elevation of temperature, while S_NiFe2O4_ and S_ZnFe2O4_ don’t change obviously.

The comparison between this calculation method and the method similar to Cr-rich oxides were shown in Figure 7a. The solubility of magnetite was measured by experiments in dilute aqueous solutions saturated with H_2_, and the results with 779 mol/kg H_2_ at 473K, 523K, and 573K were labelled with hollow squares, circles, and triangles [27,29,30]. Although the calculation results in this study lack corresponding experiments to prove the accuracy, and the assumed equilibrium was slightly different, the ferrous ions were similar. Therefore, according to the comparison, the experimental data points in the references were near the calculated curves at high temperature, and it can be judged that the calculated method is feasible.

In addition, the solubility was close to Ref. [20] (shown in Figure 7c) in the pH range of PWR primary circuit environment, and the minimum solubility of ZnFe_2_O_4_ is close to 10^−13^ mol/L, which confirms the high stability of this spinel oxide at high temperatures. In consequence, according to the lower solubility of ZnCr_2_O_4_ and ZnFe_2_O_4_ than other oxides, it has verified that zinc injection technology enhances the corrosion resistance and stability of alloys used in PWR primary circuit

### 3.3. Al-Rich Oxide

The calculation of solubility of ZnAl_2_O_4_, FeAl_2_O_4_, and NiAl_2_O_4_ is similar to Section 3.2. It is assumed that Al_2_O_3_ and MAl_2_O_4_ (M represents Zn, Fe, and Ni) coexist in the solution [26]. The possible dissolution equilibrium reactions are presumed in Table 10. The sum of ions solubility in aqueous solution composed the solubility of spinel oxides (Equations (29)–(31)), and the data is plotted in Figure 8 with the increase in temperature and pH.



(29)
$$S_{{\mathrm{FeAl}}_{2}O_{4}}=\left[{\mathrm{Fe}}^{2+}\right]+\left[\mathrm{Fe}{\left(\mathrm{OH}\right)}^{+}\right]+\left[\mathrm{Fe}{\left(\mathrm{OH}\right)}_{2}^{0}\right]+\left[{\mathrm{Fe}(\mathrm{OH})}_{3}^{-}\right]+\left[{\mathrm{Fe}(\mathrm{OH})}_{4}^{2-}\right]$$


(30)
$$S_{{\mathrm{NiAl}}_{2}O_{4}}=\left[{\mathrm{Ni}}^{2+}\right]+\left[\mathrm{Ni}{\left(\mathrm{OH}\right)}^{+}\right]+\left[\mathrm{Ni}{\left(\mathrm{OH}\right)}_{2}^{0}\right]+\left[{\mathrm{Ni}(\mathrm{OH})}_{3}^{-}\right]+\left[{\mathrm{Ni}(\mathrm{OH})}_{4}^{2-}\right]$$


(31)
$$S_{{\mathrm{ZnAl}}_{2}O_{4}}=\left[{\mathrm{Zn}}^{2+}\right]+\left[\mathrm{Zn}{\left(\mathrm{OH}\right)}^{+}\right]+\left[\mathrm{Zn}{\left(\mathrm{OH}\right)}_{2}^{0}\right]+\left[{\mathrm{Zn}(\mathrm{OH})}_{3}^{-}\right]+\left[{\mathrm{Zn}(\mathrm{OH})}_{4}^{2-}\right]$$



Given that the value of S_ZnAl2O4_ is extremely small at high temperatures, the logarithm of S_ZnAl2O4_ is calculated in Figure 8c–f. The lowest solubility of ZnAl_2_O_4_ among all spinel oxides demonstrates the highest stability in PWR primary circuit at high temperature. After zinc-aluminium is simultaneously injected into PWR coolant, the replacements of Fe^2+^, Ni^2+^, Cr^3+^ by Zn^2+^ and Al^3+^ react between the surface of alloys and the solution. ZnAl_2_O_4_, as the main product, with extremely low solubility and Gibbs free energy changes of Reaction (16) proves the positive effects of this new method on the improvement of corrosion resistance of structural materials. The effects of ZAWC on different materials are in difference, and the mechanism of the oxide film generation process is studied in the next chapter.

## 4. Mechanism Discussion

### 4.1. Thermodynamic Property

Figure 9 compares the solubility of various substances in the oxide film, formed on different alloys at a typical temperature of PWR primary circuit (573 K), and a typical water chemistry pH range (pH_300°C_ = 6.9~7.3).

Carbon steel is normally oxidized to form various iron oxides under PWR primary circuit water chemistry. When zinc is injected, the oxide (Fe_3_O_4_) reacts to form ZnFe_2_O_4_, which forms the spinel oxide on the surface of carbon steel. The solubility reduces dramatically from 10^−7^ mol/L (Fe_3_O_4_) to 10^−11^ mol/L (ZnFe_2_O_4_), which improves the corrosion resistance, and the spinel oxide film becomes more stable. When ZAWC is applied, ZnAl_2_O_4_ forms on the surface of carbon steel, and the solubility reduces sharply to 10^−413^ mol/L. Approximately 95% of the Fe in carbon steel (shown in Table 1) is considered to have the tendency to generate zinc ferrite and zinc aluminate, which directly enhance the corrosion resistance and stability.

Compared with carbon steel, the content of Cr and Ni in stainless steel increases to about 18% and 12%, respectively. However, Fe is still the main element. A double-layer Fe_3_O_4_-FeCr_2_O_4_ film forms on the surface under normal water chemistry. The outer layer is porous, composed of high iron and low nickel, and shows the properties of n-type semiconductor. The inner layer oxide is rich in chromium, relatively dense, and less porous, thus presenting p-type semiconductor properties. The corrosion mechanism of stainless steel in high temperature water, proposed by J. Robertson, demonstrated that a duplex layered oxide was formed with the inner layer, growing by the centre of water along oxide micropores, and the outer layer, growing by the diffusion of metal ions. The corrosion rate was controlled by the solid state diffusion of Fe ions along grain boundaries in the oxide layer. The relatively fast diffusion of Fe and Ni through the oxide to the outer layer leads to the preferential dissolution of the steel. The outer layer forms at the oxide surface and is precipitated to relieve the supersaturation from the surrounding dissolution of metal ions [31].

The high temperature water environment with zinc injection promotes the formation of a thin and stable ZnFe_2_O_4_-ZnCr_2_O_4_ double layer film on the surface of stainless steel. The earlier zinc is added, the thinner the oxide films and the higher relative content of ZnCr_2_O_4_ in the film. The solubility of inner oxide decreases from 10^−6^ mol/L (FeCr_2_O_4_) to 10^−9.7^ mol/L (ZnCr_2_O_4_). It has been pointed out that, in the initial stage of oxidation, the growth rate of oxides on the surface of materials was relatively high. After long-term soaking in coolant, the growth rate decreased, and the stability of oxide films increased. The formation of a dense Zn_x_Fe_1−x_Cr_2_O_4_ inner layer is the main reason for the corrosion resistance of component materials under ZWC condition. The replacement of Fe^2+^ and Ni^2+^ by Zn^2+^ can react with the inner and outer layers of oxides and, at the same time, Zn^2+^ enter the oxide films.

When zinc and aluminium are simultaneously injected into PWR coolant, it is shown in Figure 5 that the Gibbs free energy changes of Reaction (16) in the inner layer is the lowest among the formation reactions of ZnAl_2_O_4_ and decreases with the increase in temperature, that is to say that the generation of ZnAl_2_O_4_ in inner oxide may react more spontaneously in PWR primary circuit conditions. The extremely low value of S_ZnAl2O4_ also improves the corrosion resistance of oxide films formed on stainless steel.

Nickel is the main element of Ni-based alloys, and ZnFe_2_O_4_-ZnCr_2_O_4_ double-layer oxide film is generated from NiFe_2_O_4_-NiCr_2_O_4_ under ZWC. The solubility decreases from 10^−10^ mol/L (NiFe_2_O_4_) to 10^−11^ mol/L (ZnFe_2_O_4_) for outer layer and 10^−3^ mol/L (NiCr_2_O_4_) to 10^−9.7^ mol/L (ZnCr_2_O_4_) in inner layer. This change in solubility is greater than FeCr_2_O_4_/ZnCr_2_O_4_, and the nickel content in Ni-based alloy is the highest among these materials. As a result, the zinc injection technology has more effective efforts on Ni-based alloys than carbon steel.

The free energy of formation of zinc-containing spinel is lower than that of corresponding zinc-free spinel. This means that the formation of ZnCr_2_O_4_, ZnFe_2_O_4_, and ZnAl_2_O_4_ are thermodynamically favoured over that of FeCr_2_O_4_, NiCr_2_O_4_, Fe_3_O_4_, NiFe_2_O_4_, FeAl_2_O_4_, and NiAl_2_O_4_ when the spinel are newly formed on the fresh metal surfaces. Based on thermodynamic calculations, ZnCr_2_O_4_ has not only a wider stable area in the potential-pH diagram but also a lower solubility under simulated PWR primary water conditions than other spinel [9,18].

### 4.2. Crystallographic Property

All the oxides formed on carbon steel, stainless steel, and Ni-based alloys discussed above are spinel oxides. The spinel structure of AB_2_O_4_ crystals was first determined in 1915 by Bragg [32] and Nishikawa [33]. The structure is face-centre cubic arrangement of oxygen ions, with metal ions occupying half of the octahedral and one-eighth of the tetrahedral interstitial sites within the anion sub lattice. It has been defined by Barth [34] that, if all the tetrahedral sites are occupied by A divalent cation ions and all the octahedral site by B trivalent ions, A[B_2_]O_4_, the structure is normal (N). On the other hand, if the tetrahedral sites are fully occupied by B ions and the octahedral sites are occupied by equal numbers of A and B ion, B[AB]O_4_, which is called inverse (I). There is also a statistical distribution called random, in which the tetrahedral sites contain 1/3 A and 2/3 B cations, and the general formula can be written as A_1−v_^n+^B_v_^m+^ [A_v_^n+^B_2−v_^m+^] O_4_^2−^.

The factors affecting the distribution of cations in spinel crystals are extremely complex, i.e., ion radius, ion charge, the configuration of electro layer, and so on. The radius of metal cation ions are compared in Table 11 and the order of radius is Al^3+^ < Ni^2+^ < Fe^3+^ < Cr^3+^ < Zn^2+^ < Fe^2+^. The lattice energy corresponding to different cations is listed in Table 12. The location of each element in the oxide is controlled by its corresponding diffusion rate in the spinel-type oxide. For stainless steel and Ni-based alloy, with zinc injection technology applied to PWR primary circuit, Cr-rich oxides are formed in the inner layer, and Cr^3+^ has the strongest octahedral lattice energy, which tends to occupy the octahedral sites. This results in Cr remaining in the inner layer due to its strong preference to occupy octahedral sites in spinel. At the same time, Zn^2+^ has the strongest tetrahedral lattice energy. Thus, diffusion rates through the oxide are slow, and ZnCr_2_O_4_ can be stable in the oxide film [35,36,37].

The types of different spinel oxides, formed on different metal materials under different simulated PWR water chemistry, are listed in Table 13. A positive SPE means the spinel is normal and the cation distribution is predicted to be more stable, while the negative value of SPE represents the inverse distribution is preferred. The value of TSE can judge the structural stability of spinel oxides. Therefore, the relatively lower TSE values of ZnAl_2_O_4_, ZnCr_2_O_4_, and ZnFe_2_O_4_ indicate the high stability of these three spinel, formed on alloys under ZWC and ZAWC conditions, which is consistent with the results analysed in Section 3.

In order to develop an understanding of why one cation arrangement is more stable than another physically, the components of the electronic contribution to the SPE can be examined, that is anion preference energy and cation preference energy. When A cation ion stabilizes the oxygen orbitals more than B cation, the oxygen preference energy will favour the normal structure, with A ions in tetrahedral sites. Otherwise, the oxygen preference energy favours the inverse spinel structure.

With Zn^2+^ and Al^3+^ in PWR coolant, the oxide formed on the surface of carbon steel are almost composed of ZnAl_2_O_4_, and in consequence, the effect of this new technology can significantly improve the stability of carbon steel. It has been reported before that the effect of ZAWC, on the improvement of corrosion resistance of Ni-based alloy, was not as obvious as that of carbon steel. Crystallographic properties may be an important basis for explaining this phenomenon. It is worth noting that the anion preference energy of FeAl_2_O_4_ and NiAl_2_O_4_ are significantly higher than ZnAl_2_O_4_, and as a result, Fe^2+^ and Ni^2+^ are easier to combine with the oxygen ions in the spinel, and the stability of bonding is higher. Due to the high content of Ni in Ni-based alloys and the Zn^2+^/Al^3+^ concentration in solution that can enter the passive film to participate in the formation of spinel is limited, the presence of Ni^2+^ prevents Zn^2+^ from reactions, and NiAl_2_O_4_ is easier to generate than ZnAl_2_O_4_. Besides, FeAl_2_O_4_ can also produce in the oxide film, but due to the low percentage of Fe, the content of FeAl_2_O_4_ is less than NiAl_2_O_4_. The TSE of NiAl_2_O_4_ is lower than ZnAl_2_O_4_, and the solubility of NiAl_2_O_4_ is just a little lower than that of NiFe_2_O_4_-NiCr_2_O_4_, which formed in the normal PWR water chemistry, so the enhancement of corrosion resistance of Ni-based alloy is not obvious.

## 5. Conclusions

We investigated the inhabitation mechanism of ZWC and ZAWC on the corrosion of structural materials in PWR primary circuit. The ${\Delta G}^{\theta }\left(T\right)$ values of the reactions of Fe^2+^, Ni^2+^ in oxide films replaced by Zn^2+^, or Fe^3+^ replaced by Al^3+^ are extremely negative, and these replace reactions are highly spontaneous. Furthermore, we observe the solubility of ZnAl_2_O_4_ in high temperature water are extremely low, which contributes to the improvement in the corrosion resistance of carbon steels and stainless steels by the zinc-aluminium simultaneous injection into PWR primary coolant. On the other hand, the values of anion preference energy of NiAl_2_O_4_ and FeAl_2_O_4_ are much larger than that of ZnAl_2_O_4_ and Fe^2+^ and Ni^2+^ are easier to combine with the oxygen ions in the spinel than Zn^2+^ in ZnAl_2_O_4_. The concentration of Al^3+^ in the coolant is limited, so NiAl_2_O_4_ is formed, preferentially, on the surface of Ni-based alloy, and its content is greater than that of ZnAl_2_O_4_. Therefore, the corrosion resistance of Nickel alloys is mainly determined by the solubility and thermodynamic properties of NiAl_2_O_4_.

## Figures and Tables

**Figure 1 entropy-24-00245-f001:**
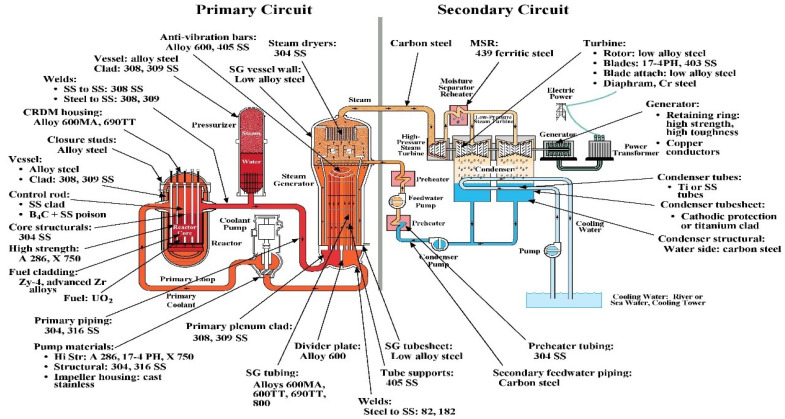
Distribution of structural materials used in PWRs [7].

**Figure 2 entropy-24-00245-f002:**
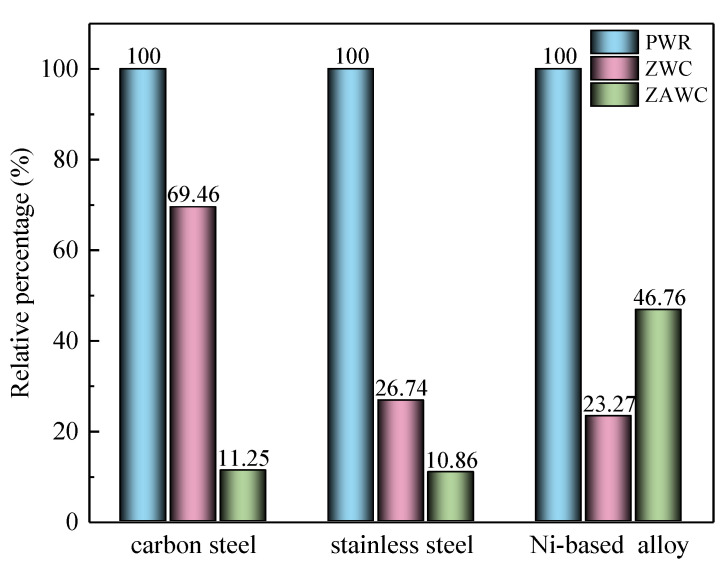
Effects of ZWC and ZAWC on the corrosion of different materials [13].

**Figure 3 entropy-24-00245-f003:**
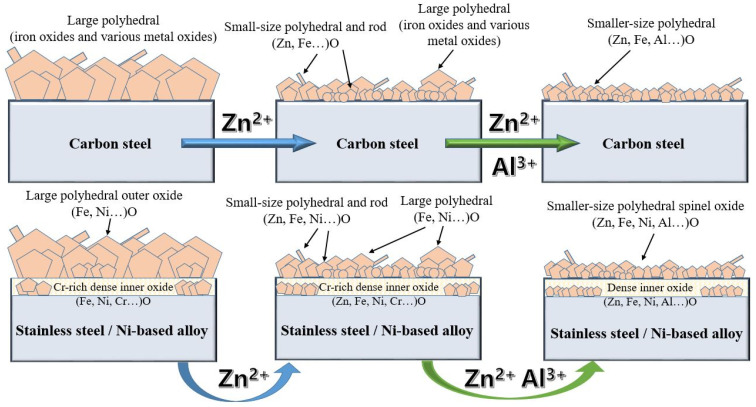
The mechanism diagram of the oxide formed on metals in PWR circuit with zinc and zinc-aluminium simultaneous injection.

**Figure 4 entropy-24-00245-f004:**
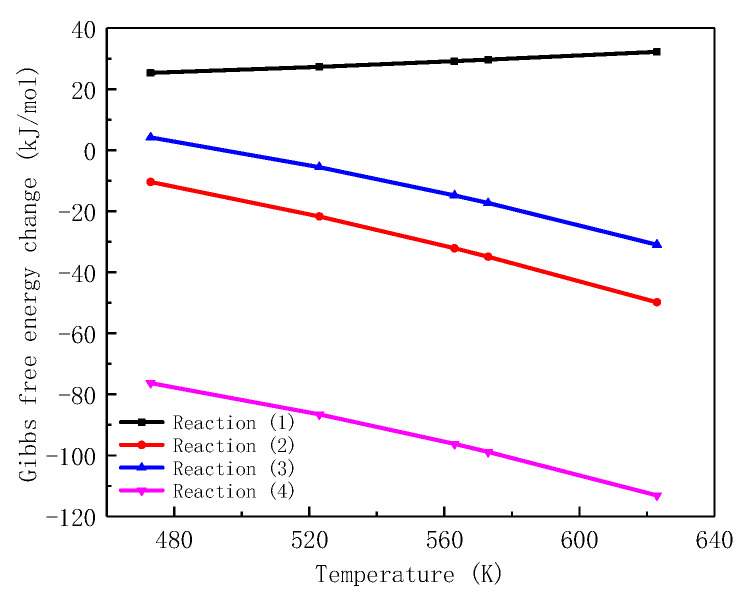
Gibbs free energy changes for ZnFe_2_O_4_ and ZnCr_2_O_4_ possible formation reactions.

**Figure 5 entropy-24-00245-f005:**
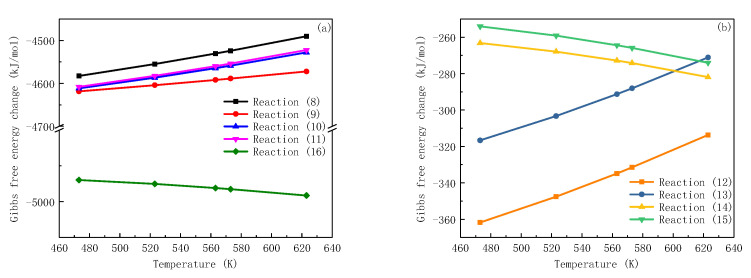
Gibbs free energy changes of possible formation reactions: (**a**) Formation of ZnAl_2_O_4_, (**b**) Formation of FeAl_2_O_4_, NiAl_2_O_4_.

**Figure 6 entropy-24-00245-f006:**
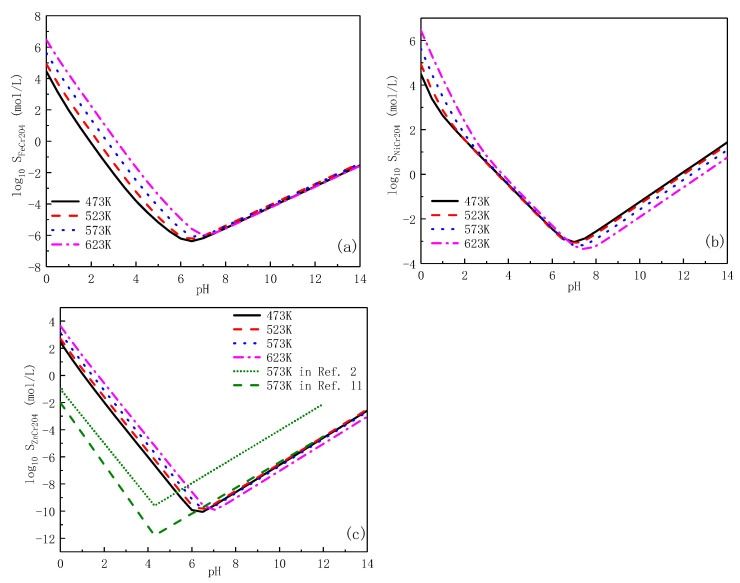
Solubility of FeCr_2_O_4_ (**a**), NiCr_2_O_4_ (**b**), and ZnCr_2_O_4_ (**c**) at different temperature as a function of pH.

**Figure 7 entropy-24-00245-f007:**
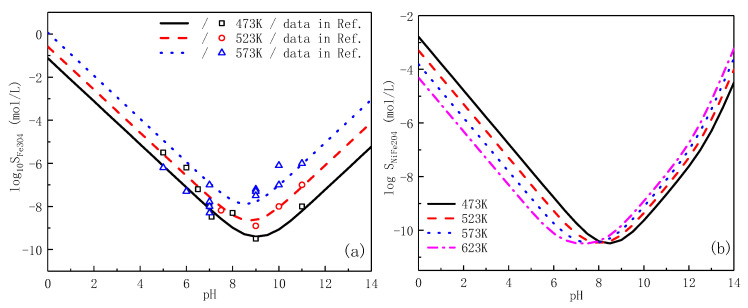
Solubility of Fe_3_O_4_ (**a**), NiFe_2_O_4_ (**b**), and ZnFe_2_O_4_ (**c**) at different temperature as a function of pH [20,27,29,30].

**Figure 8 entropy-24-00245-f008:**
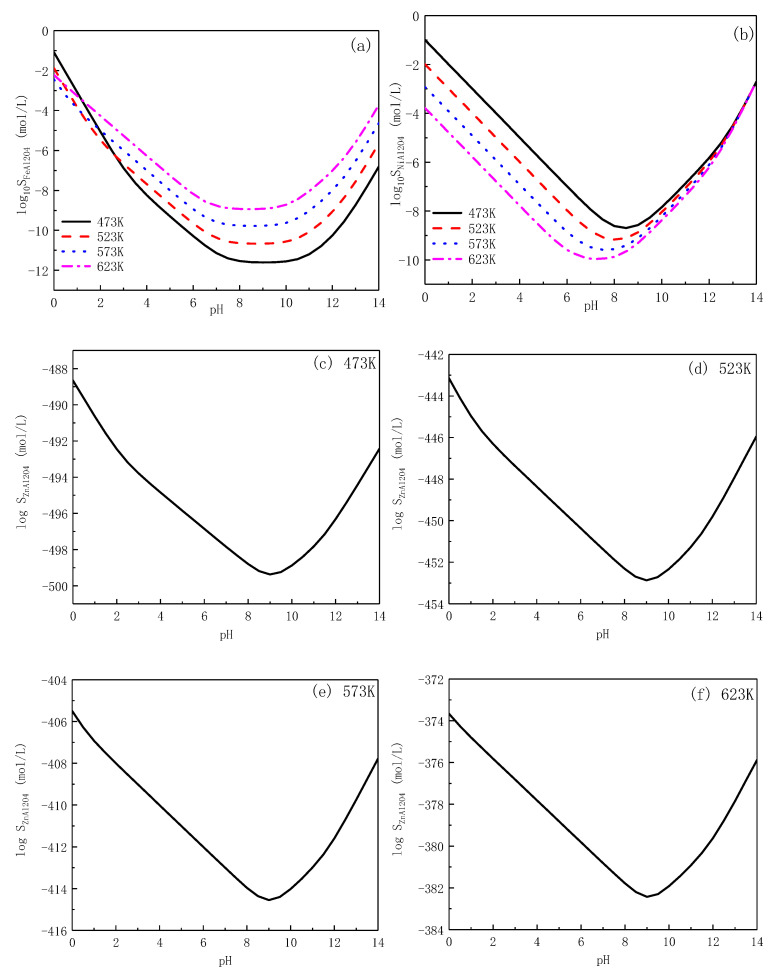
Solubility of FeAl_2_O_4_ (**a**), NiAl_2_O_4_ (**b**), and ZnAl_2_O_4_ (**c**–**f**) at different temperature as a function of pH.

**Figure 9 entropy-24-00245-f009:**
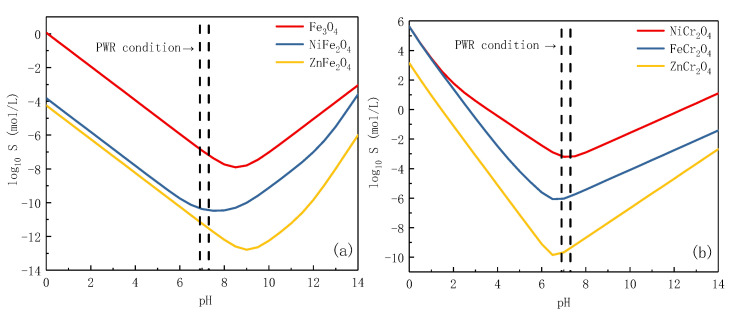
Comparison of solubility of oxides at PWR primary circuit condition: (**a**) Outer layer, (**b**) Inner layer.

**Table 1 entropy-24-00245-t001:** Component materials in PWR primary system (%).

	Composition (%)				
	C	Cr	Ni	Fe	Mn
A106B CS	≤0.3	≤0.4	≤0.4	Bal.	0.29–1.06
A508-3 CS	≤0.22	≤0.25	0.5–0.80	Bal.	1.15–1.6
304L SS	≤0.03	18–20	8–12	Bal.	≤2.0
316L SS	≤0.03	16–18	10–14	Bal.	≤2.0
Incoloy 800	≤0.1	19–23	30–35	37–47	≤1.5
Inconel 600	≤0.15	14–17	72	6–10	≤1.0
Inconel 690	0.023	30.39	60	8.88	0.23

**Table 2 entropy-24-00245-t002:** Thermodynamic data of species used in formation equations [2,11].

Species	${\Delta G}_{f}^{\circ }\left(298K\right)$ (kJ·mol^−1^)	$S_{298}^{{}^{\circ}}$ (J·mol^−1^·K^−1^)	A	B (×10^−3^)	C (×10^5^)
Fe_3_O_4_	−1015.45	146.4	91.5	201	-
NiFe_2_O_4_	−974.6	125.9	77.4	235	1.42
FeCr_2_O_4_	−1347.02	147	163.2	22.36	−31.95
NiCr_2_O_4_	−1269.14	129.7	167.2	17.87	−21.05
ZnFe_2_O_4_	−1063.5	151.67	161.5	28.93	−26.53
ZnCr_2_O_4_	−1434.0	116.3	167.4	14.23	−25.1
Zn^2+^	−147.1	−156.5	−164	-	-
Fe^2+^	−78.9	−182.1	−188	-	-
Ni^2+^	−48.2	−170.7	269.6	-	-

**Table 3 entropy-24-00245-t003:** Calculated ${\Delta G}_{f}^{\circ }\left(T\right)$ at different temperatures (kJ/mol).

Species	${\Delta G}_{f}^{\circ }\left(473K\right)$	${\Delta G}_{f}^{\circ }\left(523K\right)$	${\Delta G}_{f}^{\circ }\left(563K\right)$	${\Delta G}_{f}^{\circ }\left(573K\right)$	${\Delta G}_{f}^{\circ }\left(623K\right)$
Fe_3_O_4_	−1048.13	−1059.81	−1069.83	−1072.43	−1085.95
NiFe_2_O_4_	−1003.65	−1014.31	−1023.53	−1025.92	−1038.47
FeCr_2_O_4_	−1379.03	−1390.21	−1399.72	−1402.18	−1414.87
NiCr_2_O_4_	−1298.62	−1309.19	−1318.24	−1320.57	−1332.70
ZnFe_2_O_4_	−1096.55	−1108.08	−1117.89	−1120.42	−1133.50
ZnCr_2_O_4_	−1460.99	−1470.79	−1479.21	−1481.39	−1492.71
Zn^2+^	−112.57	−100.54	−90.35	−87.72	−74.19
Fe^2+^	−38.85	−24.92	−13.13	−10.09	5.56
Ni^2+^	−30.10	−28.49	−28.13	−28.16	−29.01

**Table 4 entropy-24-00245-t004:** Calculated ${\Delta G}_{f}^{\circ }\left(T\right)$ at different temperatures (kJ/mol) [2,11,22,23,24,25].

Species	${\Delta G}_{f}^{\circ }\left(298K\right)$ (kJ·mol^−1^)	$S_{298}^{{}^{\circ}}$ (J·mol^−1^·K^−1^)	A	B (×10^−3^)	C (×10^5^)
ZnAl_2_O_4_	−6671	87.03	166.52	15.48	−46.02
FeAl_2_O_4_	−1879.67	106.299	123.544	-	-
NiAl_2_O_4_	−1791.12	98.324	131.567	-	-
Fe^3+^	−15.4	−382.5	−204	-	-
Al^3+^	−485.3	−325	113.115	−0.506	-
Cr^3+^	−215.48	−370.3	488.7	-	-
Fe_2_O_3_	−740.99	89.96	98.28	77.82	−14.85
ZnO	−321.9	43.16	47.58	3.93	−7.504

**Table 5 entropy-24-00245-t005:** Calculated ${\Delta G}_{f}^{\circ }\left(T\right)$ at different temperatures (kJ/mol).

Species	${\Delta G}_{f}^{\circ }\left(473K\right)$	${\Delta G}_{f}^{\circ }\left(523K\right)$	${\Delta G}_{f}^{\circ }\left(563K\right)$	${\Delta G}_{f}^{\circ }\left(573K\right)$	${\Delta G}_{f}^{\circ }\left(623K\right)$
ZnAl_2_O_4_	−6692.04	−6699.99	−6706.89	−6708.69	−6718.10
FeAl_2_O_4_	−1903.65	−1912.13	−1919.35	−1921.21	−1930.82
NiAl_2_O_4_	−1814.05	−1822.34	−1829.43	−1831.27	−1840.76
Fe^3+^	60.42	84.78	104.97	110.11	136.34
Al^3+^	−433.34	−419.99	−409.70	−407.17	−394.85
Cr^3+^	−171.95	−165.97	−162.88	−162.33	−160.83
Fe_2_O_3_	−761.66	−769.19	−775.67	−777.36	−786.13
ZnO	−331.31	−334.59	−337.38	−338.10	−341.81

**Table 6 entropy-24-00245-t006:** Possible reactions of dissolution process.

Oxidation Product	Dissolution Equilibrium Reactions
FeCr_2_O_4_	$\left(1\right){\mathrm{FeCr}}_{2}O_{4}{+2H}^{+}\to {\mathrm{Fe}}^{2+}{+\mathrm{Cr}}_{2}O_{3}{+H}_{2}O$ $\left(2\right){\mathrm{FeCr}}_{2}O_{4}{+4H}^{+}{+2H}_{2}\to {\mathrm{Fe}+2\mathrm{Cr}}^{2+}{+4H}_{2}O$ $\left(3\right){\mathrm{FeCr}}_{2}O_{4}{+H}_{2}\to {\mathrm{Fe}+2\mathrm{CrO}}_{2}^{-}{+2H}^{+}$ $\left(4\right){\mathrm{FeO}+2\mathrm{CrO}}_{2}^{-}{+2H}^{+}\to {\mathrm{FeCr}}_{2}O_{4}{+H}_{2}O$ $\left(5\right){\mathrm{Fe}}_{2}O_{3}{+4\mathrm{CrO}}_{2}^{-}{+4H}^{+}{+H}_{2}\to {2\mathrm{FeCr}}_{2}O_{4}{+3H}_{2}O$ $\left(6\right){\mathrm{Cr}}_{2}O_{3}{+6H}^{+}\to {2\mathrm{Cr}}^{3+}{+3H}_{2}O$ $\left(7\right){\mathrm{Cr}}_{2}O_{3}{+4H}^{+}{+H}_{2}\to {2\mathrm{Cr}}^{2+}{+3H}_{2}O$ $\left(8\right){\mathrm{HFeO}}_{2}^{-}{+H}^{+}\to {\mathrm{FeO}+H}_{2}O$
NiCr_2_O_4_	$\left(1\right){\mathrm{NiCr}}_{2}O_{4}{+2H}^{+}\to {\mathrm{Cr}}_{2}O_{3}{+\mathrm{Ni}}^{2+}{+H}_{2}O$ $\left(2\right){\mathrm{NiO}+2\mathrm{CrO}}_{2}^{-}{+2H}^{+}\to {\mathrm{NiCr}}_{2}O_{4}{+H}_{2}O$ $\left(3\right){\mathrm{NiO}+2\mathrm{CrO}}_{4}^{2-}{+4H}^{+}{+3H}_{2}\to {\mathrm{NiCr}}_{2}O_{4}{+5H}_{2}O$ $\left(4\right){\mathrm{Cr}}_{2}O_{3}{+6H}^{+}\to {2\mathrm{Cr}}^{3+}{+3H}_{2}O$ $\left(5\right){\mathrm{Cr}}_{2}O_{3}{+4H}^{+}{+H}_{2}\to {2\mathrm{Cr}}^{2+}{+3H}_{2}O$ $\left(6\right){\mathrm{HNiO}}_{2}^{-}{+H}^{+}\to {\mathrm{NiO}+H}_{2}O$ $\left(7\right){\mathrm{NiCr}}_{2}O_{4}{+H}_{2}\to {2\mathrm{CrO}}_{2}^{-}{+2H}^{+}+\mathrm{Ni}$
ZnCr_2_O_4_	$\left(1\right){\mathrm{ZnCr}}_{2}O_{4}{+8H}^{+}\to {\mathrm{Zn}}^{2+}{+2\mathrm{Cr}}^{3+}{+4H}_{2}O$ $\left(2\right){\mathrm{ZnCr}}_{2}O_{4}{+6H}^{+}{+H}_{2}\to {\mathrm{Zn}}^{2+}{+2\mathrm{Cr}}^{2+}{+4H}_{2}O$ $\left(3\right){\mathrm{ZnCr}}_{2}O_{4}{+4H}^{+}{+2H}_{2}\to {\mathrm{Zn}}^{2+}{+2\mathrm{Cr}}^{2+}{+4H}_{2}O$ $\left(4\right){\mathrm{ZnCr}}_{2}O_{4}{+H}_{2}\to {\mathrm{Zn}+2\mathrm{CrO}}_{2}^{-}{+2H}^{+}$ $\left(5\right){\mathrm{HZnO}}_{2}^{-}{+2\mathrm{CrO}}_{2}^{-}{+3H}^{+}\to {\mathrm{ZnCr}}_{2}O_{4}{+2H}_{2}O$ $\left(6\right){\mathrm{HZnO}}_{2}^{-}{+2\mathrm{CrO}}_{4}^{2-}{+5H}^{+}{+3H}_{2}\to {\mathrm{ZnCr}}_{2}O_{4}{+6H}_{2}O$ $\left(7\right){\mathrm{ZnOH}}^{+}{+2\mathrm{CrO}}_{4}^{2-}{+3H}^{+}{+3H}_{2}\to {\mathrm{ZnCr}}_{2}O_{4}{+5H}_{2}O$

**Table 7 entropy-24-00245-t007:** Thermodynamic data of species used in dissolution equations [2,11].

Species	${\Delta G}_{f}^{\circ }\left(298K\right)$ (kJ·mol^−1^)	$S_{298}^{{}^{\circ}}$ (J·mol^−1^·K^−1^)	A	B (×10^−3^)	C (×10^5^)
H^+^	0	−22.2	−71	-	-
H_2_	0	130.6	27.28	3.264	0.502
H_2_O	−237.19	70.08	75.44	-	-
Fe	0	27.15	14.1	29.7	1.799
FeO	−246.35	79.5	48.79	8.37	−2.803
HFeO_2_^−^	−379.18	41.92	−508.1	-	-
Cr_2_O_3_	−1046.84	81.17	119.3	9.096	−15.64
Cr^2+^	−176.15	−120.9	314.5	-	-
CrO_2_^−^	−535.93	117.3	−386.7	-	-
CrO_4_^2−^	−736.8	80.33	−474	-	-
Ni	0	30.12	16.99	29.46	-
NiO	−215.94	37.99	−20.88	157.2	16.28
HNiO_2_^−^	−349.22	62.84	−409.7	-	-
Zn	0	41.63	22.38	10.04	-
HZnO_2_^−^	−464	62.84	−409.6	-	-
ZnOH^+^	−329.28	−50.21	265.2	-	-

**Table 8 entropy-24-00245-t008:** Possible reactions of dissolution process.

Oxidation Product	Dissolution Equilibrium Reacuions
${\mathrm{Fe}}_{3}O_{4}$	$\left(1\right){\mathrm{Fe}}_{3}O_{4}{+2H}^{+}\to {\mathrm{Fe}}_{2}O_{3}{+\mathrm{Fe}}^{2+}{+H}_{2}O$ $\left(2\right){\mathrm{Fe}}_{3}O_{4}{+H}^{+}\to {\mathrm{Fe}}_{2}O_{3}+\mathrm{Fe}{\left(\mathrm{OH}\right)}^{+}$ $\left(3\right){\mathrm{Fe}}_{3}O_{4}{+H}_{2}O\to {\mathrm{Fe}}_{2}O_{3}+\mathrm{Fe}{\left(\mathrm{OH}\right)}_{2}^{0}$ $\left(4\right){\mathrm{Fe}}_{3}O_{4}{+2H}_{2}O\to {\mathrm{Fe}}_{2}O_{3}+\mathrm{Fe}{\left(\mathrm{OH}\right)}_{3}^{-}{+H}^{+}$
${\mathrm{NiFe}}_{2}O_{4}$	$\left(1\right){\mathrm{NiFe}}_{2}O_{4}{+2H}^{+}\to {2\mathrm{Fe}}_{2}O_{3}{+\mathrm{Ni}}^{2+}{+H}_{2}O$ $\left(2\right){\mathrm{NiFe}}_{2}O_{4}{+H}^{+}\to {2\mathrm{Fe}}_{2}O_{3}+\mathrm{Ni}{\left(\mathrm{OH}\right)}^{+}$ $\left(3\right){\mathrm{NiFe}}_{2}O_{4}{+H}_{2}O\to {\mathrm{Fe}}_{2}O_{3}+\mathrm{Ni}{\left(\mathrm{OH}\right)}_{2}^{0}$ $\left(4\right){\mathrm{NiFe}}_{2}O_{4}{+2H}_{2}O\to {\mathrm{Fe}}_{2}O_{3}+\mathrm{Ni}{\left(\mathrm{OH}\right)}_{3}^{-}{+H}^{+}$ $\left(5\right){\mathrm{NiFe}}_{2}O_{4}{+3H}_{2}O\to {\mathrm{Fe}}_{2}O_{3}+\mathrm{Ni}{\left(\mathrm{OH}\right)}_{4}^{2-}{+2H}^{+}$
${\mathrm{ZnFe}}_{2}O_{4}$	$\left(1\right){\mathrm{ZnFe}}_{2}O_{4}{+2H}^{+}{+H}_{2}O\to 2\mathrm{Fe}{\left(\mathrm{OH}\right)}_{3}{+\mathrm{Zn}}^{2+}$ $\left(2\right){\mathrm{ZnFe}}_{2}O_{4}{+H}^{+}{+3H}_{2}O\to 2\mathrm{Fe}{\left(\mathrm{OH}\right)}_{3}+\mathrm{Zn}{\left(\mathrm{OH}\right)}^{+}$ $\left(3\right){\mathrm{ZnFe}}_{2}O_{4}{+4H}_{2}O\to 2\mathrm{Fe}{\left(\mathrm{OH}\right)}_{3}+\mathrm{Zn}{\left(\mathrm{OH}\right)}_{2}^{0}$ $\left(4\right){\mathrm{ZnFe}}_{2}O_{4}{+5H}_{2}O\to 2\mathrm{Fe}{\left(\mathrm{OH}\right)}_{3}+\mathrm{Zn}{\left(\mathrm{OH}\right)}_{3}^{-}{+H}^{+}$ $\left(5\right){\mathrm{ZnFe}}_{2}O_{4}{+6H}_{2}O\to 2\mathrm{Fe}{\left(\mathrm{OH}\right)}_{3}+\mathrm{Zn}{\left(\mathrm{OH}\right)}_{4}^{2-}{+2H}^{+}$

**Table 9 entropy-24-00245-t009:** Thermodynamic data of species used in dissolution equations [27,28].

Species	${\Delta G}_{f}^{\circ }\left(298K\right)$ (kJ·mol^−1^)	$S_{298}^{{}^{\circ}}$ (J·mol^−1^·K^−1^)	A	B (×10^−3^)	C (×10^5^)
Fe(OH)^+^	−270.8	−120	450	-	-
Fe(OH)_2_	−447.43	−80	435	-	-
Fe(OH)_3_^−^	−612.65	−70	560	-	-
Ni(OH)^+^	−227.2	−49.7	−200	-	-
Ni(OH)_2_	−406	−71	100	-	-
Ni(OH)_3_^−^	−586.5	−133	300	-	-
Ni(OH)_4_^2−^	−743.7	−252	460	-	-
Fe(OH)_3_	−705.29	106.7	127.61	41.639	−42.17
Zn(OH)^+^	−339.7	62.76	41.84	-	-
Zn(OH)_2_^0^	−519.27	61.55	33.47	-	-
Zn(OH)_3_^−^	−700.44	2.98	159.83	-	-
Zn(OH)_4_^2−^	−864.69	−27.51	89.54	-	-

**Table 10 entropy-24-00245-t010:** Possible reactions of dissolution process.

Oxidation Product	Dissolution Equilibrium Reactions
MAl_2_O_4_ (M: Zn, Fe, Ni)	$\left(1\right){\mathrm{MAl}}_{2}O_{4}{+2H}^{+}\to {\mathrm{Al}}_{2}O_{3}{+M}^{2+}{+H}_{2}O$ $\left(2\right){\mathrm{MAl}}_{2}O_{4}{+H}^{+}\to {\mathrm{Al}}_{2}O_{3}+M{\left(\mathrm{OH}\right)}^{+}$ $\left(3\right){\mathrm{MAl}}_{2}O_{4}{+H}_{2}O\to {\mathrm{Al}}_{2}O_{3}+M{\left(\mathrm{OH}\right)}_{2}^{0}$ $\left(4\right){\mathrm{MAl}}_{2}O_{4}{+2H}_{2}O\to {\mathrm{Al}}_{2}O_{3}+M{\left(\mathrm{OH}\right)}_{3}^{-}{+H}^{+}$ $\left(5\right){\mathrm{MAl}}_{2}O_{4}{+3H}_{2}O\to {\mathrm{Al}}_{2}O_{3}+M{\left(\mathrm{OH}\right)}_{4}^{2-}{+2H}^{+}$

**Table 11 entropy-24-00245-t011:** Radius of metal cation ions (Å).

M^2+^	Mg	Ni	Co	Zn	Fe	Mn
Radius	0.65	0.72	0.74	0.74	0.76	0.80
M^3+^	Al	Ni	Co	Fe	Mn	Cr
Radius	0.50	0.62	0.63	0.64	0.66	0.69

**Table 12 entropy-24-00245-t012:** Lattice energy of metal cation ions.

Cation Ion	Lattice Energy of Octahedral Sites (kcal/mol)	Preferred Lattice
Cr^3+^	16.6	Octahedral (strongest)
Ni^2+^	9.0	Tetrahedral
Fe^2+^	−9.9	Tetrahedral
Fe^3+^	−13.3	Tetrahedral
Zn^2+^	−31.6	Tetrahedral (strongest)

**Table 13 entropy-24-00245-t013:** Crystallographic parameters of spinel oxides (eV).

Spinel	Type	Cation Distribution (tet, oct, oct)	Total Structure Energy (TSE)	Anion Preference Energy	Cation Preference Energy	Structure Preference Energy (SPE)
FeAl_2_O_4_	N	Fe^2+^Al^3+^Al^3+^	−590.965	4.11	−0.82	2.06
NiAl_2_O_4_	I	Al^3+^Ni^2+^Al^3+^	−619.534	4.23	−3.3	−1.43
ZnAl_2_O_4_	N	Zn^2+^Al^3+^Al^3+^	−682.214	−0.04	2.45	1.76
FeCr_2_O_4_	N	Fe^2+^Cr^3+^Cr^3+^	−638.243	1.67	−1.99	3.7
NiCr_2_O_4_	N	Ni^2+^Cr^3+^Cr^3+^	−665.768	2.18	−4.5	0.61
ZnCr_2_O_4_	N	Zn^2+^Cr^3+^Cr^3+^	−730.68	−2.04	1.29	3.84
Fe_3_O_4_	I	Fe^3+^Fe^2+^Fe^3+^	−683.899	0	−0.14	−0.86
NiFe_2_O_4_	I	Fe^3+^Ni^2+^Fe^3+^	−714.186	0.24	−2.67	−4.23
ZnFe_2_O_4_	N	Zn^2+^Fe^3+^Fe^3+^	−775.186	−2.23	−3.14	0.77

## Data Availability

The data presented in this study are openly available in [ResearchGate] at [10.5573/ieek.5013.50.1.034], reference number [2]; in [Elsevier] at [10.1016/j.corrsci.2011.06.011] [10.1016/S0016-7037(96)00339-0] [10.1016/S0010-938X(96)00067-4], reference number [11,25,27]; in [Springer] at [10.1023/A:1021866025627] [10.1007/BF00645517], reference number [28,29].

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
