# Peer review of "Thermodynamic Analysis and Crystallographic Properties of MFe2O4, MCr2O4 and MAl2O4 (M = Fe, Ni, Zn) Formed on Structural Materials in Pressurized Water Reactor Primary Circuit under Zinc and Zinc-aluminum Water Chemistry"

_entropy, 2022, doi:10.3390/e24020245_

Round 1
Reviewer 1 Report
First of all I would like to thank you for considering our journal to publish your manuscript. I have read your manuscript entitled: Thermodynamic Analysis and Crystallographic Properties of MFe2O4, MCr2O4 and MAl2O4 (M=Fe, Ni, Zn) Formed on Structural Materials in PWR Primary Circuit under ZWC and ZAWC with great interest. The paper contains material which is worthy of publication highlighting the effectiveness of zinc and zinc-aluminum injection on corrosion behavior of carbon steels, stainless steels and Ni- based alloys.
The paper is very clear and well written, the inserted references are appropriate, although the subsections are very hard to follow and understand, therefore the manuscript needs only some minor revision before it can be published, presented below point-by-point:
- Please change the title in order to not use abbreviations which were not detailed (ex. PWR, ZWC and ZAWC)
- The research value, significance and future work should be described in the final stage of the abstract.
- PWR should be defined as first appear in abstract page 1 line 12 instead of page 1 line 33.
- Aims need to be clearly and concisely stated and added at the end of introduction. Not only what was done/investigated, but why?
- Remove the SPE description on page 16 line 385.
- I suppose that references 1, 7, 10, 14, 15, 31 Thesis. Please write accordingly to journal instructions: Author 1, A.B. Title of Thesis. Level of Thesis, Degree-Granting University, Location of University, Date of Completion (https://www.mdpi.com/journal/entropy/instructions)
- The homogeneity of the reference section needs to be done most of the references are abbreviated, but some aren`t (2, 20). Journal Corrosion is having no abbreviation (ex. 13, 21). Write again reference 32, there are some mistakes. Please check and revise!
Based on these, I advise the authors to rectify the above mentioned errors and we hope to re-evaluate the revised manuscript.
Reviewer 2 Report
This manuscript deals mainly with the strength of Zn-Al simultaneous injection in corrosion resistance of different types of alloyed steel materials. By calculating the thermodynamic parameter of the series of reactions, finding the balance between the chemical driving force and solubility to conclude the corrosion resistance efficiency. This paper is of interest, but certain improvements are necessary before publication.
1. The introduction must be well directed and well-referenced: This work belongs to quite an established field, while the first two paragraphs are not referenced at all. Although keywords such as "it has been reported", "It was well known that…" are used to describe the problem, I couldn't find a guide for general readers. Especially the reference regarding the inner and outer layers is essential for a broad readership.
2. Figure 3 carries the main message, and further progress of the manuscript is loosely explained. Before starting section 2.1, figure 3 should be nicely explained. Some structural and morphological characterization and schematics would be clear to pin the argument.
3. Conclusion part is quite problematic. This part starts with a very long sentence, "in order to…… calculated in this research." The authors must consider splitting into small parts, each conveying a message. The very following sentence is, 'it can be concluded…" sounds weak. The author should avoid using extensive past tense/passive voice in the conclusion part (in many other sections, similar modifications are possible). Instead, "we conclude", or "we observe" should be used.
Minor:
1. 13 tables are presented in the manuscript, some of which are also presented in figures. Do the authors feel it necessary to keep all the tables in the main manuscript? Or a more competent representation is possible?
2. In many places, the 5 or 6 digit parameters are shown in tables, which can be converted to scientific digits.
3. Missing: Unit for the radius in Table 11.
*************** end of report **************
Round 2
Reviewer 1 Report
The author has made a substantial improvement for this article. The manuscript can be accepted for publication in the present form.
Reviewer 2 Report
The revised manuscript provides more clear presentation in the introduction, paragraph transition, and conclusion. It can be considered for publication.